# Impact of the SARS-CoV-2 Virus Pandemic on Patients with Bronchiectasis: A Multicenter Study

**DOI:** 10.3390/antibiotics11081096

**Published:** 2022-08-12

**Authors:** Adrián Martínez-Vergara, Rosa Mª Girón Moreno, Casilda Olveira, María Victoria Girón, Adrián Peláez, Julio Ancochea, Grace Oscullo, Miguel Ángel Martínez-García

**Affiliations:** 1Instituto de Investigación Sanitaria La Princesa, Hospital Universitario La Princesa, 28006 Madrid, Spain; 2Facultad de Medicina, Universidad Autónoma de Madrid, 28049 Madrid, Spain; 3Hospital Regional Universitario de Málaga, 29010 Malaga, Spain; 4Instituto de Investigación Biomédica de Málaga (IBIMA), Universidad de Málaga, 29016 Malaga, Spain; 5CIBER de Enfermedades Respiratorias, ISCIII, 28029 Madrid, Spain; 6Hospital Universitario y Politécnico La Fe, 46026 Valencia, Spain

**Keywords:** bronchiectasis, exacerbation, COVID-19, SARS-CoV-2, pandemic, E-FACED, *Pseudomonas aeruginosa*, *Haemophilus influenza*, BSI

## Abstract

Background: Infection by SARS-CoV-2 has unquestionably had an impact on the health of patients with chronic respiratory airway diseases, such as COPD and asthma, but little information is available about its impact on patients with bronchiectasis. The objective of the present study was to analyze the effect of the SARS-CoV-2 pandemic on the state of health, characteristics, and clinical severity (including the number and severity of exacerbations) of patients with non-cystic fibrosis bronchiectasis. Methods: This study was multicenter, observational, and ambispective (with data collected before and during the SARS-CoV-2 pandemic), and included 150 patients diagnosed with non-cystic fibrosis bronchiectasis. Results: A significant drop was observed in the number and severity of the exacerbations (57% in all exacerbations and 50% in severe exacerbations) in the E-FACED and BSI multidimensional scores, in the pandemic, compared with the pre-pandemic period. There was also a drop in the percentage of sputum samples positive for pathogenic microorganisms in general (from 58% to 44.7%) and, more specifically, *Pseudomonas aeruginosa* (from 23.3% to 13.3%) and *Haemophilus influenzae* (from 21.3% to 14%). Conclusions: During the SARS-CoV-2 period, a significant reduction was observed in the exacerbations, severity, and isolations of pathogenic microorganisms in patients with bronchiectasis.

## 1. Introduction

Since the appearance of the first cases of pneumonia due to SARS-CoV-2 in December 2019, millions of people from all over the world have been infected, many of them with severe cases [1,2,3] and significant consequences [4,5] or death [6,7,8] as a result. On 30 January 2020, the World Health Organization (OMS) declared the disease now known as the novel coronavirus 2019 (COVID-19) a worldwide public health emergency [9]. Since then, some countries have experienced a varying number of waves of greater or lesser severity, depending on the protective measures adopted, the infectiousness, the characteristics of the individuals affected [10,11,12], and the aggressiveness of the various strains of the virus that have appeared, as well as the vaccination coverage achieved [13]. Various social and preventive protection measures were imposed according to the geographical region (mask wearing, social distancing, and even—in periods of high infection—lockdown) while the various vaccines currently on the market were still being developed. This situation led to changes in the healthcare of patients with chronic respiratory diseases [14,15,16,17,18,19,20], with delays in both diagnosis and therapy [21], as well as difficulties in accessing health centers, due not only to the restrictions imposed but also to the fear of being infected among the population [22].

Furthermore, the number of infectious exacerbations produced by respiratory viruses (found in more than 50% of cases) [23,24] also reduced in both healthy individuals and, above all, patients with chronic airway diseases, as reported in patients with chronic obstructive pulmonary disease (COPD) [25,26]. One obvious illustration of this trend was the virtual disappearance of the seasonal influenza virus [27] during the first waves of the pandemic, which has been primarily attributed to respiratory protection measures and social distancing. 

In this context, it is necessary to analyze how the pandemic has influenced patients with bronchiectasis, considering that it represents the third most frequent chronic inflammatory disease of the airway (after asthma and COPD), with a high social and economic impact and an important implication upon the quality of life of these patients [28,29]. However, the data on the relationship between the pandemic caused by SARS-CoV-2 and the control of, and impact on, bronchiectasis (BE) patients are scarce. In this respect, only one single-center study comparing the pre-pandemic period and the pandemic itself [30], initially designed for the validation of a questionnaire on the quality of life in BE, has found a decrease of up to 50% in the number of exacerbations; the authors mainly attributed this reduction to favorable environmental factors (less environmental pollution and widespread respiratory protection) during the pandemic. Nevertheless, this development did not translate into changes in the symptoms or quality of life of patients in a stable phase. Since BE is acquiring ever greater epidemiological significance [31] and exacerbations are crucially linked to the severity, quality of life, prognosis, and even costs associated with this disease [28], the results obtained by Crichton et al. [30] are hugely important. 

Our working hypothesis was that of a possible decrease in the number and severity of exacerbations, as well as a variable impact on other circumstances affecting these patients. Our objective was thus to analyze the impact of infection by SARS-CoV-2 during the pandemic, compared with the pre-pandemic period, on the number and severity of exacerbations and on other variables of interest, such as clinical, microbiological, therapeutic, lung-function, and analytical aspects. 

## 2. Materials and Methods

### 2.1. Study Design

The study was observational, ambispective, and multicenter, with the participation of three public tertiary Spanish centers (not-for-profit and with free access) with a specialized BE unit, including patients of varying severity, meeting the referral criteria described by the Spanish Society of Pneumology and Thoracic Surgery (SEPAR) [32]. All the included patients were informed about the characteristics of the study and gave their informed consent. The project was approved by the Ethics Committee of the Hospital La Princesa (registry no. 3992). The study’s methodology follows the recommendations established for observational studies: STROBE (http://www.strobe-statement.org/).

### 2.2. Criteria for Inclusion and Exclusion

The criteria for inclusion were: a diagnosis of non-cystic fibrosis bronchiectasis (NCFBE) demonstrated by a high resolution computerized tomography (HRCT) chest scan, following the criteria of Naidich [33], with a compatible clinical picture [34]; adult status (18 years or more); and follow-up by the BE unit of one of the participating centers. The criteria for exclusion were: a change of residence to another Spanish Autonomous Community (resulting in a loss of follow-up) or refusal to participate in the study.

Initially, 173 consecutive patients diagnosed with BE were included, although 23 were later excluded (Figure 1). Finally, 150 patients completed the study (50 from each of the three participating centers). 

### 2.3. Follow-Up

Considering 14 March 2020 (declaration of the state of emergency) as day 0 of the SARS-CoV-2 pandemic in Spain [35], two follow-up periods were established: one prior to the pandemic (15 March 2019–15 March 2020) and the other during the pandemic period (16 March 2020–15 March 2021). All the included patients underwent a complete follow-up in both phases of the study (pre-pandemic and pandemic) via appointments every 3–6 months, according to a clinician’s criteria, based on the severity presented by the patient. 

Due to the restrictions imposed by the pandemic, some of the appointments during the pandemic period were conducted by telephone. Accordingly, all the patients had a telephone contact number for fulfilling the designated medical consultations and for providing information about exacerbation episodes, in case they could not attend in person.

### 2.4. Analyzed Variables

#### 2.4.1. Baseline Variables

The following baseline variables were analyzed: age, gender, body mass index (BMI), smoking habit, age at diagnosis of BE, etiology, number of lobes affected in the HRCT, uni- or bilateral distribution, and morphology (cylindrical or cystic), as well as the Charlson [36] and BACI (Bronchiectasis Aetiology Comorbidity Index) indexes [37]. 

#### 2.4.2. Variables Compared between the Pre-Pandemic and Pandemic Periods 

The following variables were recorded at the first appointment in the pre-pandemic period and the last in the pandemic, for the purposes of comparison: the quantity and type of expectoration, according to Murray’s classification [38]; the degree of dyspnea on the mMRC (modified Medical Research Council) scale [39]; lung function studies (FVC and FEV_1_); changes in usual treatment; analytical data: leukocytes, neutrophils, eosinophils, erythrocyte sedimentation rate (ESR), fibrinogen, albumin, C-reactive protein (CRP); the FACED [40] and e-FACED [23] scores; the BSI (Bronchiectasis Severity Index) score [41]; and the HADS (Hospital Anxiety and Depression Scale) [42] and CAT (COPD Assessment Test) questionnaires [43]. Cut-offs of 11 or less points in the HADS-A or HADS-D questionnaires [44] and more than 10 in the CAT symptoms questionnaire were considered pathological.

The following data were recorded at all the appointments undertaken in both periods: the number and severity of the exacerbations, the number of oral and intravenous antibiotic courses, the total number of days of antibiotic therapy, and infection (or not) by the SARS-CoV-2 virus.

An exacerbation [45] was defined as a situation in which a patient presented a worsening of three or more of the following symptoms in the previous 48 h: cough, volume and/or consistency of sputum, purulence of sputum, dyspnea and/or intolerance of exercise, asthenia and/or general malaise, and hemoptysis, as well as a need for a change in treatment and the exclusion of other causes of clinical deterioration.

An exacerbation was considered mild–moderate when it required the prescription of treatment by the oral route and severe when it required intravenous antibiotic therapy and/or hospitalization. 

Moreover, the following microbiological data were recorded at each appointment: the percentage of valid sputum samples (>25 polymorphonuclear and <10 epithelial cells) [46], the percentage of sputum samples positive for a potentially pathogenic microorganism (PPM) [47], and the percentage of sputum samples positive for *Pseudomonas aeruginosa* (PA), *Haemophilus influenzae* (HI), and *Staphylococcus aureus* (SA). Culture results were expressed as CFU per milliliter and ≥10^3^ CFU were considered positive for a given PPM.

In the case of death, this was confirmed by the corresponding official death certificate.

### 2.5. Statistical Analysis

The results were presented as the mean ± standard deviation, median and interquartile range (IQR), or absolute number and percentage, depending on whether the variables were quantitative or qualitative and whether or not they followed a normal distribution, as verified by the Kolmogorov–Smirnov test. 

Both parametrical (Student’s t-test for repeated measurements) and non-parametrical (Wilcoxon) tests were used to compare the quantitative variables recorded during the pre-pandemic and pandemic periods. In the case of qualitative variables, proportions were compared by means of the chi-square test, as well as Fisher’s exact test, where necessary. The statistical package IBM SPSS Statistics for Windows, Version 20.0. Armonk, NY was used. 

In the absence of any supporting data in the literature at the start of the study, we established: the main variable as a reduction by at least one exacerbation per patient per year between the pre-pandemic and pandemic period; an approximation of loss of patients of 10%; an alpha error of 0.05%; and a statistical power of 20% for repeated measurements, making the number of patients needed to perform the analyses 150. In order to achieve the greatest representation possible for all three centers taking part in the study, each was required to include 50 valid, consecutive patients for analysis before the study could be considered complete. We used PASS 2022, the Power analysis and sample size program (NCSS Statistical Software, Kaysville, UT, USA) as a sample size software.

## 3. Results 

### 3.1. Baseline Characteristics of the Patients

As can be seen in Table 1, of the 150 patients who were finally included, 115 (76.7%) were women with a mean age of 61.3 ± 16.1 years. The Charlson Index was 2.9 ± 1.9 and the most frequent etiology was post-infection (49.3%). The mean FACED and BSI scores were 1.6 ± 1.4 and 5.4 ± 3.6, respectively. The percentage of FEV_1_ was 75.6% ± 25 and the median of exacerbations was 1 (IQR: 1–2).

### 3.2. Comparison between the Pre-Pandemic and Pandemic Periods

The mean number of days of follow-up in the pre-pandemic period was 357 ± 35 days, whilst it was 361 ± 37 days in the pandemic, with no significant differences between the two (*p* = 0.99). Three patients died during the follow-up. Twelve patients (8.7%) presented infection by SARS-CoV-2 but none of them required hospitalization. The median of in-person appointments was 3 (IQR: 2–4) and 2 (IQR: 1–2), *p* = 0.001, and that of telephone appointments was 0 (IQR: 0–1) and 2 (IQR: 2–3), *p* < 0.001, in the pre-pandemic and pandemic periods, respectively.

#### 3.2.1. Exacerbations

In the pre-pandemic period, the median of the total number of exacerbations was 1 (IQR: 1–2) and the mean was 1.4 ± 1.2, while during the pandemic, the median was 0 (IQR: 0–1) and the mean was 0.6 ± 0.8, which corresponded to *p* < 0.0001 and *p* < 0.001, respectively (Figure 2). This represented a 57% drop in the total number of exacerbations.

The proportion of patients without any exacerbations thus rose from 22.6% in the pre-pandemic period to 63.1% in the pandemic (*p* < 0.001) (Figure 3). Furthermore, the percentage of patients with one exacerbation fell from 37.3% to 22.1% (*p* = 0.012), while that of patients with two exacerbations fell from 23.3% to 10.7% (*p* = 0.031), that of patients with three fell from 12.6% to 4% (*p* = 0.039), and that of those with more than three went from 4% to 0% (*p* = 0.042).

Figure 3 shows the percentage of patients who experienced 0, 1, 2, 3, or >3 exacerbations in the pre-pandemic period (red) and the pandemic period (blue).

Furthermore, as can be seen in Table 2, the median of the number of courses of oral antibiotic therapy in the pre-pandemic period went from 1 (IQR: 1–2) to 0 (IQR: 0–1) in the pandemic (*p* < 0.0001), while the median of the days of oral antibiotic therapy dropped from 14 (IQR: 6–23,25) to 0 (IQR: 0–14) (*p* < 0.0001) and the mean fell from 16.9 ± 16 to 7.3 ± 12.5 (*p* < 0.001). Similarly, the percentage of patients who experienced at least one severe exacerbation fell from 13.5% to 6.8% (*p* = 0.013), representing a 50% reduction. Moreover, the mean of the days of intravenous antibiotic therapy decreased from 1.96 ± 5.44 to 0.83 ± 3.64 (*p* = 0.009).

#### 3.2.2. Microbiological Aspects

The median of the sputum samples collected before the pandemic was 2 (IQR: 1–3), while during the pandemic it was 1 (IQR: 0–3); *p* < 0.01. Of these, the means of sputum samples positive for PPM were, respectively, 1 (0–3) and 1 (0–2); *p* = 0.02. The percentage of patients from whom it was possible to collect valid sputum samples was greater in the pre-pandemic period, with a mean (range) of 2.5 (0–9), as opposed to 1.2 (0–6) during the pandemic; *p* = 0.012. As can be seen in Table 3, during the pandemic there was a 13.3% reduction, in those patients capable of expectorating, in the percentage of positive sputum samples for any PPM other than SA, including HI (7.3%) and PA (10%). 

#### 3.2.3. Severity Scores 

The mean score of the E-FACED scale fell from 1.8 ± 1.5 to 1.43 ± 1.43 (*p* = 0.030) (Figure 4) and that of the BSI fell from 5.4 ± 3.6 to 4.4 ± 3.3 (*p* = 0.047) (Figure 5). There was also a drop in the FACED score, but this was not statistically significant (Table 4).

#### 3.2.4. Quality of Life and Psychomorbidity

Table 4 shows that there were no differences in the CAT scores, although the percentage of patients with clear symptoms of anxiety or depression (cut-off point > 11 in the HADS questionnaire) decreased during the pandemic compared with the pre-pandemic period.

#### 3.2.5. Clinical, Functional, Analytical, and Therapeutic Aspects

As regards the clinical manifestations, a comparison between the two periods did not reveal any statistically significant differences in the degree of dyspnea, according to the mMRC scale, or the presence of expectoration, or any differences in lung function, analytical parameters, or the usual treatment (Table 5). 

## 4. Discussion

According to our results, there was a reduction in the general severity of BE during the pandemic (compared with the pre-pandemic period), due to a very significant drop in both the number and severity of exacerbations and in their treatments, along with a decrease in the isolations of PPM, including PA and HI. No changes were observed in clinical, analytical, functional, or therapeutic aspects, or in the quality of life, despite a decrease in the percentage of patients with symptoms of anxiety and depression. 

Regarding the decrease in the number and severity of exacerbations, our results are very similar to those obtained in other studies of patients with BE or COPD. In this respect, Crichton ML et al. [30] observed a reduction in the mean of exacerbations from 2.01 in the period 2019–2020 to 1.12 in 2020–2021, with an increase in the percentage of patients free from exacerbations from 25.6% to 52.3%; there was also a decrease in the corresponding courses of systemic antibiotic therapy. Similarly, a meta-analysis by Alqahtani et al. [25] found a 50% reduction in hospitalizations due to exacerbations of COPD during the pandemic. These results can probably be explained by measures adopted during the pandemic period investigated in study (16 March 2020–15 March 2021) in Spain, such as the mandatory use of a mask and hand disinfection, lockdown (15 March 2020–12 June 2020), curfew, shop closures, social distancing and restriction of massive events, leading to a decrease in viral infections, which cause a considerable percentage of exacerbations in BE patients. [48]. These results are hugely significant in terms of both epidemiology and clinical application, since it has already been proven that both the number and the severity of exacerbations have a negative impact on the natural history of BE patients [48,49,50,51]. Moreover, the data provided could have implications in the management of these patients, and by extension to patients with other chronic respiratory diseases, in favor of the recommendation to use mask in crowded places, visits to hospitals, or other healthcare centers, and promotion of teleconsultation and domiciliary treatments, with the purpose to avoid overexposure to respiratory germs.

From the microbiological viewpoint, although the quantity of patients presenting a valid sputum sample was significantly lower in the pandemic period, there was also a very noteworthy drop in the percentage of sputum samples positive for PPM, particularly HI and PA—a finding closely related to severity and exacerbations in BE [28,29,49,50,52,53,54,55]—even though no differences were observed in the baseline treatments between the two time periods. Once again, respiratory protection and social distancing that impede the transmission of viruses, along with fewer exacerbations, prevented a subsequent growth of pathogenic bacteria (bacterial superinfection), thereby increasing the time free from exacerbation and probably helping the immune system to achieve control over chronic bacterial infection or eradicate PPM from the bronchial mucosa [56]. Another explanation may be that the “threshold” quantity of bacteria allowing detection in culture is exceed more often in the pre-pandemic era due to more frequent exacerbations, however we were not able to analyze this hypothesis since we have not information about quantitative cultures in samples from an exacerbation phase and stable condition. 

Another noteworthy aspect is the lack of improvement in the quality of life and the parameters of anxiety and depression (factors that are normally closely linked) that would be expected to accompany a decrease in the severity of BE [44]. Any slight impact on the quality of life and psychomorbidity could be limited, however, by the negative impact of the lack of improvement in baseline symptoms, in a context of fear of contagion, reduced social contacts, and increased loneliness in the pandemic, particularly during lockdown. Nevertheless, not surprisingly, the severity scores for BE that include exacerbations and the microbiological pattern, such as BSI and E-FACED, significantly improved, while FACED does not have a sufficient discriminatory capacity as it does not include the number and severity of exacerbations.

Among the strengths of this study, it should be noted that it is the first to be designed to evaluate the effect of the pandemic on BE patients on a multidimensional basis (measuring both objective and subjective parameters), as well as being multicentered, with participation from hospitals from the center, east, and south of Spain. Among the limitations of this study, it is important to point out some of them: firstly, the follow-up of the patients was necessarily conducted in a different way as a consequence of the pandemic, with a significant reduction in the number of in-person appointments and collections of sputum samples during that period. Accordingly, the analyses were carried out with data related to the percentage of total sputum samples collected (in both exacerbation stable phase) and samples positive for PPM in both periods, rather than with absolute data; secondly, we have no data on quantitative cultures; thirdly, all three centers included had specialized BE clinics so we cannot rule out a severity bias towards more severe cases, and finally, we cannot rule out an underestimate of exacerbations, especially in their milder forms, in the times of less direct contact with the patients, although a continuous, on-demand telephone service was established in every case to minimize this circumstance.

## 5. Conclusions

In conclusion, the number and severity of exacerbations and the degree of infection by PPM dropped significantly in BE patients during the SAR-CoV-2 pandemic compared with the period prior to the pandemic. Although no causal relationship could be established, it is highly probable that these results are linked to the increase in respiratory protection measures and social distancing that would impede contagion, especially from viruses. Controlled, prospective studies are needed to confirm whether the implementation of such measures is—as the present study suggests—effective in preventing exacerbations in BE patients. 

## Figures and Tables

**Figure 1 antibiotics-11-01096-f001:**
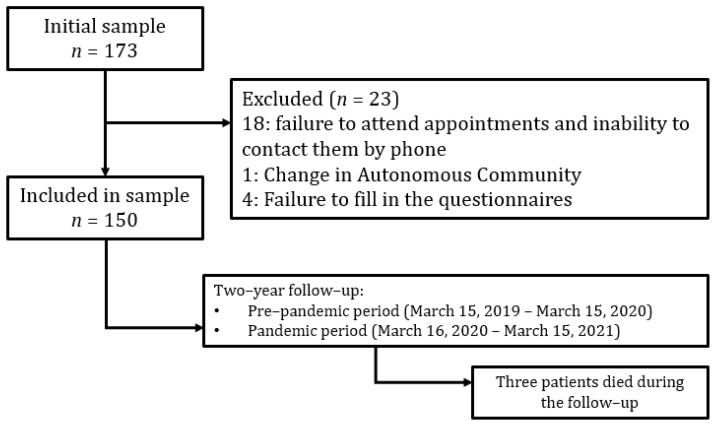
Flowchart of the study.

**Figure 2 antibiotics-11-01096-f002:**
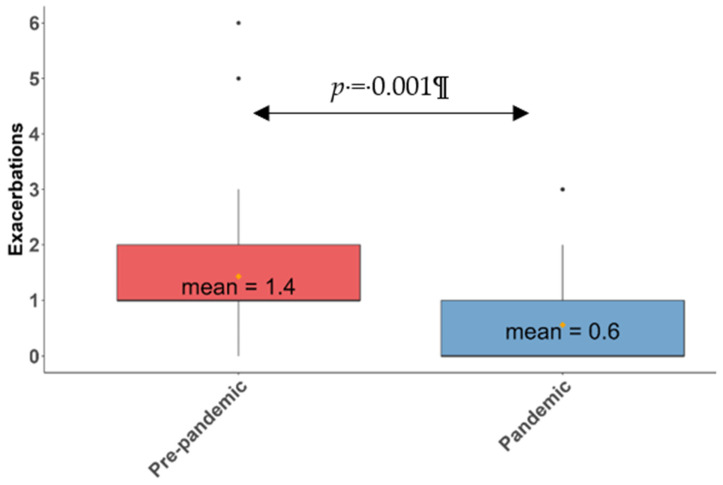
Box plot. Mean and median of exacerbations in the pre-pandemic and pandemic periods. The thick black line represents the median, and the yellow dot represents the mean.

**Figure 3 antibiotics-11-01096-f003:**
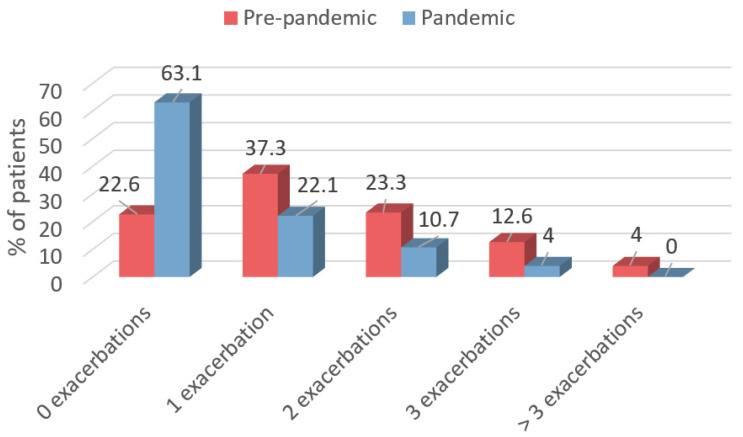
Percentage of patients who experienced 0, 1, 2, 3, or >3 exacerbations in the pre-pandemic and pandemic periods.

**Figure 4 antibiotics-11-01096-f004:**
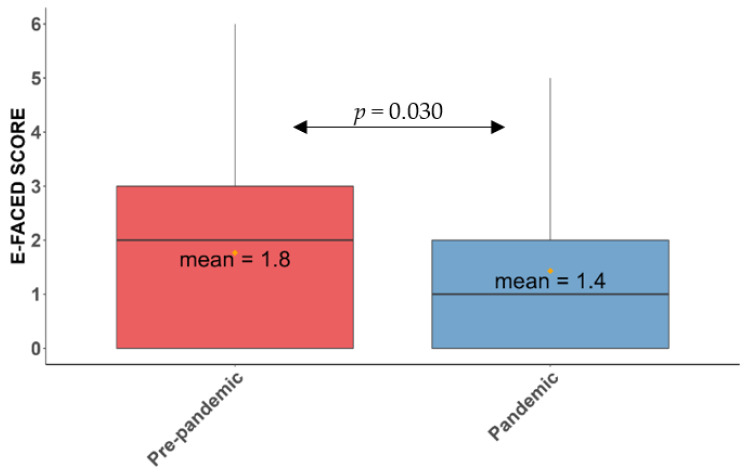
Mean score of the E-FACED scale in the pre-pandemic and pandemic periods. The yellow dots represent the mean of the E-FACED in the pre-pandemic (red) and pandemic (blue) periods.

**Figure 5 antibiotics-11-01096-f005:**
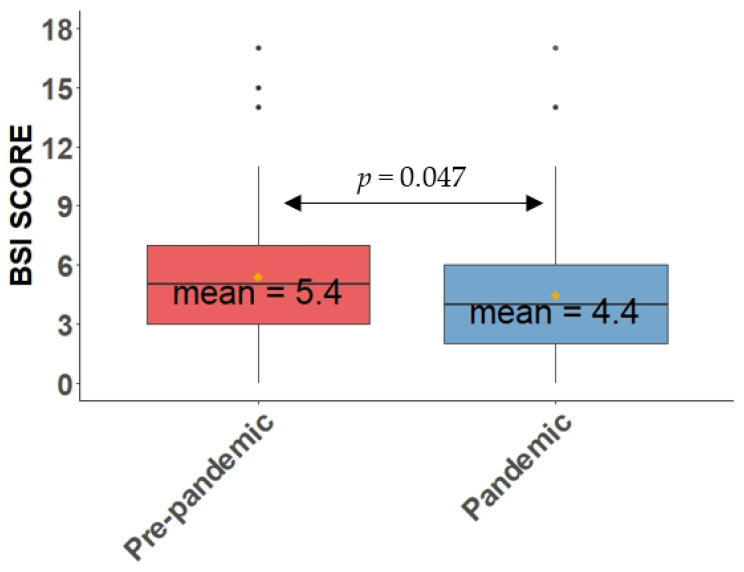
Mean score of the BSI scale in the pre-pandemic and pandemic periods. The yellow dots represent the mean of the BSI in the pre-pandemic (red) and pandemic (blue) periods.

**Table 1 antibiotics-11-01096-t001:** Baseline characteristics of the patients.

Gender Women Men	(76.7%) (23.3%)
Age, years	61.3 ± 16.1
BMI (kg/m^2^)	25.3 ± 4.6
Smoking Never Active smoker Ex-smoker	93 (62%) 14 (9.3%) 43 (28.17%)
Packs-per-year index	12.6 ± 17.7
Age at diagnosis of BE, years	50 ± 19.5
Etiology Post-infection Unknown Immunodeficiencies Genetic diseases Ciliary dyskinesia Diseases of the connective tissue COPD Congenital malformations Intestinal inflammatory diseases Bronchial asthma Hematological diseases Hyperimmune response Bronchiolitis obliterans Others	74 (49.3%) 17 (11.4%) 13 (8.7%) 10 (6.7%) 9 (6%) 6 (4%) 5 (3.3%) 3 (2%) 3 (2%) 2 (1.3%) 2 (1.3%) 2 (1.3%) 1 (0.7%) 3 (2%)
Number of pulmonary lobes affected 1 2 3 >3	28 (18.7%) 56 (37.3%) 19 (12.7%) 47 (31.4%)
Distribution Unilateral Bilateral	48 (32%) 102 (68%)
Cystic bronchiectasis	15 (10%)
Charlson Index	2.9 ± 1.9
BACI Index	1.66 ± 2.16
FACED	1.6 ± 1.4
E-FACED	1.8 ± 1.5
BSI	5.4 ± 3.6
FEV_1_ (mL) FEV1_1_ (%) FEV_1_/FVC	1861.8 ± 726.5 75.6 ± 25 79.2 ± 17.5
Exacerbations Number of oral antibiotic courses Number of intravenous antibiotic courses Days of oral antibiotic therapy Days of oral intravenous therapy	1 (1–2) 1 (1–2) 0 (0–0) 14 (6–23.25) 0 (0–0)

BMI—body mass index; BE—bronchiectasis; BACI—Bronchiectasis Aetiology Comorbidity Index; FEV_1_—forced expiratory volume during the first second.

**Table 2 antibiotics-11-01096-t002:** Antibiotic courses and accumulated days of oral and intravenous antibiotic therapy.

	Pre-Pandemic	Pandemic	*p* Value
Courses of oral antibiotic therapy	1 (1–2)	0 (0–1)	<0.0001
Courses of intravenous antibiotic therapy	0 (0–0)	0 (0–0)	<0.0001
Days of oral antibiotic therapy Median (IQR) Mean ± SD	14 (6–23,25) 16.9 ± 16	0 (0–14) 7.3 ± 12.5	<0.0001 <0.001
Severe exacerbations Median (IQR) Mean ± SD	0 (0–0) 1.96 ± 5.44	0 (0–0) 0.83 ± 3.64	<0.0001 *p* = 0.009

IQR—interquartile range; SD—standard deviation. Paired t-student test was used to compare variables with normal distribution whereas the Wilcoxon test was used in the case of non-normal distribution of the variables.

**Table 3 antibiotics-11-01096-t003:** Percentage of positive sputum samples for potentially pathogenic microorganisms during the pre-pandemic and pandemic periods.

	Pre-Pandemic *n* (%)	Pandemic *n* (%)	*p* Value
% patients with at least 1 valid sputum sample	75.3%	46.7%	0.001
Percentage of sputum samples positive for PPM	58%	44.7%	0.012
Percentage of sputum samples positive for PA	23.3%	13.3%	0.013
Percentage of sputum samples positive for HI	21.3%	14%	0.042
Percentage of sputum samples positive for SA	6%	8.7%	0.672

PPM—potentially pathogenic microorganisms; PA—*Pseudomonas aeruginosa*; HI—*Haemophilus influenzae*; SA—*Staphylococcus aureus*. Chi-square test was used for comparison between groups.

**Table 4 antibiotics-11-01096-t004:** Comparison of the FACED, E-FACED, and BSI scales and the CAT and HADS questionnaires during the pre-pandemic and pandemic periods.

	Pre-Pandemic Period	Pandemic Period	*p* Value
FACED (m ± SD)	1.6 ± 1.4	1.41 ± 1.4	0.065
E-FACED (m ± SD)	1.8 ± 1.5	1.43 ± 1.43	0.030
BSI (m ± SD)	5.4 ± 3.6	4.4 ± 3.3	0.047
CAT (m ± SD) >10 points	12.8 ± 7.59 56%	11.8 ± 6.47 52.7%	0.519 0.812
HADS-A (m ± SD) >11 points	6.47 ± 4.54 13.3%	5.49 ± 4 8%	0.084 0.059
HADS-D (m ± SD) >11 points	4.78 ± 4.1 8%	4.54 ± 3.73 2.7%	0.877 0.011

BSI—Bronchiectasis Severity Index; CAT—COPD Assessment Test; HADS-A: Hospital Anxiety and Depression Scale—Anxiety; HADS-D: Hospital Anxiety and Depression Scale—Depression—Paired *t*-student test was used to compare variables with normal distribution whereas the Wilcoxon test was used in the case of non-normal distribution of the variables. Chi-squared test was used to compare percentages.

**Table 5 antibiotics-11-01096-t005:** Comparison of the symptoms, lung function, analytical parameters, and usual treatment in the pre-pandemic and pandemic periods.

	Pre-Pandemic	Pandemic	*p* Value
Dyspnea mMRC *n* (%) Score of 0 Score of 1 Score of 2 Score of 3 Score of 4	80 (54.1%) 42 (28.4%) 19 (12.8%) 7 (4.7%) 0 (0%)	77 (52%) 44 (29.7%) 24 (16.2%) 3 (2%) 0 (0%)	0.515
Expectoration *n* (%) Present Absent	112 (75.7%) 36 (24.3%)	114 (77%) 34 (23%)	0.891 0.872
Lung function (m ± SD) FVC (% pred.) FEV_1_ (% pred.) FEV_1_/FVC	80.4 (22.5) 73.7 ± 24.3 81.1 ± 17.3	78.4 (19) 72.8 ± 21.6 83.4 ± 19.6	0.785 0.610 0.713
Analytical parameters (m ± SD) Leucocytes (mil/mm^3^) Neutrophils (%) Eosinophils (%) CPR (mg/dL) ESR Albumin (g/dL)	7070.4 ± 1722 55.8 ± 10.3 2.89 ± 2.2 0.35 ± 0.74 14.75 ± 12.5 4.17 ± 0.45	7139.8 ± 2043 56.4 ± 11.4 2.89 ± 3.07 0.44 ± 0.89 13.14 ± 12.6 4.2 ± 0.46	0.657 0.453 0.966 0.112 0.564 0.819
Treatment in stable phase (%) SABA SAMA LABA LAMA IC Nebulized bronchodilator Montelukast Mucolytics Oral corticoid Oral antibiotic Nebulized antibiotic Nebulized saline LTOT	47.3% 15% 58% 26% 52% 8.2% 1.4% 3.4% 0.7% 24.3% 21% 23% 2.7%	45.3% 18% 57% 29% 53% 9.5% 2% 6.1% 2.7% 26.3% 22.3% 20.3% 4.7%	0.699 0.546 0.901 0.739 0.934 0.645 0.766 0.332 0.211 0.867 0.876 0.624 0.423

mMRC—modified Medical Research Council; SABA—short-acting β-agonist; SAMA—short-acting muscarinic antagonist; LABA—long-acting β-agonist; LAMA—long-acting muscarinic antagonist; IC—inhaled corticoid; LTOT—long-term oxygen therapy. Paired t-student test was used to compare variables with normal distribution whereas the Wilcoxon test was used in the case of non-normal distribution of the variables. Chi-squared test was used to compare percentages.

## Data Availability

The data presented in this study are available on request from the corresponding author. The data are not publicly available.

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
