# Peer review of "Impact of the SARS-CoV-2 Virus Pandemic on Patients with Bronchiectasis: A Multicenter Study"

_antibiotics, 2022, doi:10.3390/antibiotics11081096_

Round 1

Reviewer 1 Report

Dear authors,

COVID-19 pandemic has for sure changed the characteristics of other diseases and especially of their infectious exacerbations. How the pandemic affected non-cystic fibrosis bronchectasis is an interesting topic. There are scarce articles in the literature about bronchectasis in the COVID-19 era.

-Abstract / Introduction: It summarises all the important info and states the aim of the study successfully.

-Methods: Well-written.

-Results: Good description of results. All info relevant and necessary.

-Discussion: Nice discussion of the results. No need for significant changes.

-Overall comments: Good english language use. Although well-written, topic of average interest.

Best regards.

Author Response

Response: The authors of the present manuscript would thanks the reviewer his/her very positive comments about our paper.

Reviewer 2 Report

Authors have prepared a well-written manuscript investigating the impact of the SARS-CoV-2 pandemic on patients with bronchiectasis in multiple centres across Spain. I have several comments for authors consideration, as below.

1)      There seems to be an excessive number of references cited, particularly in Introduction section. In the first sentence (Lines 34-36), for example, there is no need to cite a total of 16 references to convey the message that SARS-CoV-2 infection may result in severe complications, sequels and potentially deaths.

2)      Lines 42-44: A brief description of social and preventive protection measures were introduced, but not in the context of study setting. What were the protection measures put in place in study setting – place, and study period (pandemic period investigated in study)? This can either be described in Introduction, Discussion or Supplementary Material to provide better context of the restrictions in the study setting as this is discussed as the possible major contributor underlying study findings.

3)      The hypothesis is clear but Introduction can be strengthened by providing more justification for this study. Is there a need to know how pandemic impacted BE? Why? What are the possible policy implications arising from this?

4)      Under Methods, 2.1: Study setting can be elaborated more – especially on the population covered by this study. Are the three centres representative of a population within a certain municipal/district? With respect to BE, are all cases seen only by tertiary centres or only specific cases (for example, only severe cases)? What about the access to these centres – are they all private for-profit or government non-for-profit centres? This will give a better understanding of what population and possibly, the specific types of BE patients covered by this study.

5)      Closely related to the previous comment above, Lines 317-320 stated that strength of study is multicentred and representative. How representative are the findings? Can the findings be generalizable to other settings, or to the national level for Spain?

6)      Lines 132-135: Not entirely clear how the sample size was derived, what was the baseline exacerbation used? It would be best to provide citation here to the formula/software used for calculation.

7)      Please also provide details on the statistical software/applications used for statistical analysis (2.5).

8)      The tables and figures in Results can be improved. For Table 1, please consider using three-line tables where the top/header row has the description, while on the left column put the variables and options, leaving the right column only for values (n. %). For Tables 2-5, where p-values were presented, please insert a footnote to state what test(s) was used to derive the p-values.

9)      There is a lack of consistency on how findings were presented – for example, 3.2 was described fully in text without tables, and certain results, for example 3.2.1 was presented in box plot figures. Also, why present both median and mean (with their respective spread, IQR and SD), when normality is checked?

10)  Discussion reads well, but can be further expanded on the implications of the findings, particularly in future policy to manage BE cases. Are there any recommendations, or implications, from the findings of this study? How are the study findings important?

Author Response

Authors have prepared a well-written manuscript investigating the impact of the SARS-CoV-2 pandemic on patients with bronchiectasis in multiple centres across Spain. I have several comments for authors consideration, as below.

1) There seems to be an excessive number of references cited, particularly in Introduction section. In the first sentence (Lines 34-36), for example, there is no need to cite a total of 16 references to convey the message that SARS-CoV-2 infection may result in severe complications, sequels and potentially deaths.

Response: Thanks for the comment. We have now substantially reduced the number of references cited in the first sentence of the introduction and line 45 of the same section.

2)      Lines 42-44: A brief description of social and preventive protection measures were introduced, but not in the context of study setting. What were the protection measures put in place in study setting – place, and study period (pandemic period investigated in study)? This can either be described in Introduction, Discussion or Supplementary Material to provide better context of the restrictions in the study setting as this is discussed as the possible major contributor underlying study findings.

Response: We have added the most important specific measures adopted in Spain during the pandemic period analyzed in this study (second paragraph of the discussion section)

3)      The hypothesis is clear but Introduction can be strengthened by providing more justification for this study. Is there a need to know how pandemic impacted BE? Why? What are the possible policy implications arising from this?

Response: Thanks for commenting this important point. As we clearly highlighted now in the third paragraph of the introduction section, in our opinion it is very important the study of the impact of the pandemic on bronchiectasis (especially on exacerbations) due to some reasons: 1. Bronchiectasis is the third more frequent chronic inflammatory airway disease with a high clinical impact in the patients; 2. Most of the costs from bronchiectasis derived from exacerbations and chronic bronchial infection; 3. Our hypothesis was that the pandemic situation could reduce the number and severity of exacerbations in bronchiectasis.  

4)      Under Methods, 2.1: Study setting can be elaborated more – especially on the population covered by this study. Are the three centres representative of a population within a certain municipal/district? With respect to BE, are all cases seen only by tertiary centres or only specific cases (for example, only severe cases)? What about the access to these centres – are they all private for-profit or government non-for-profit centres? This will give a better understanding of what population and possibly, the specific types of BE patients covered by this study.

Response: We clarify this aspect in the first paragraph of the section 2.1 and discussion (limitations). All three centers are tertiary public centers with specialized BE units, one from the center, one from the east and one from the south of Spain. All centers followed the Spanish guidelines for diagnosis and management of bronchiectasis. However, as the reviewer suggested, we are aware that these are specialized centers so we cannot ensure a severity bias towards more severe cases compared with other smaller centers. Therefore, we have added new information about it, and we have add as a limitation the possibility of a severity bias.

5)      Closely related to the previous comment above, Lines 317-320 stated that strength of study is multicentred and representative. How representative are the findings? Can the findings be generalizable to other settings, or to the national level for Spain?

Response: Please see the previous response. In order to create confusion we have delete the word “representative”

6)      Lines 132-135: Not entirely clear how the sample size was derived, what was the baseline exacerbation used? It would be best to provide citation here to the formula/software used for calculation.

Response. The problem with the sample size calculation if that at the moment of the conception of the study there was any study aimed to analyze the same topic, since the study of Crichton et al was published in 2021 (ref 30). Therefore to calculate a sample size we took according to the authors´ criteria and the hypothesis of the study that a decrease in at least 1 exacerbation/patient compared the pre-pandemic and pandemic periods would be clinically significant. In fact, “a posteriori” results of the Crichton´s study concluded a reduction in the mean of exacerbations from 2.01 to 1.1 (this is approximately 1 exacerbation/year), although we did not know this data at the beginning of our study. Remarkably we achieved similar results.

On the other hand the main variable was not the number of exacerbation in the pre-pandemic period but the difference between the pre-pandemic and pandemic period. Regarding the software to calculate the sample size: “PASS 2022, Power analysis and sample size program; NCSS Statistical Software, Kaysville, Utah, USA”

7)      Please also provide details on the statistical software/applications used for statistical analysis (2.5).

Response: The statistical package SPSS Inc. 20 was used.

8)      The tables and figures in Results can be improved. For Table 1, please consider using three-line tables where the top/header row has the description, while on the left column put the variables and options, leaving the right column only for values (n. %). For Tables 2-5, where p-values were presented, please insert a footnote to state what test(s) was used to derive the p-values.

Response: We have modified the table 1 as the reviewer requested. Although all the statistical tests used has been specified in the statistical analysis section, we have added at a footnote in tables from 2 to 5 the tests used to obtain the p values.

9)      There is a lack of consistency on how findings were presented – for example, 3.2 was described fully in text without tables, and certain results, for example 3.2.1 was presented in box plot figures. Also, why present both median and mean (with their respective spread, IQR and SD), when normality is checked?

Response:  Thanks for the comments. We have used finally 5 tables and 5 figures (total 10) in our paper. Therefore we have decided to select those tables/figures that we considered more significant for the readers and left in the text the rest of the data.

Regarding the use of mean and median, we finally decided to use both measures and analyses. Although it is true the number and severity of exacerbations and other variables such as days of oral antibiotic therapy did not follow a normal distribution (and then the median and interquartile range should be used),  some papers on exacerbations in bronchiectasis also use the mean values (standard deviation). This is the case of the only paper on the impact of the pandemic on bronchiectasis exacerbation (Crichton et al did not use the median for calculated the exacerbations). So, we have decided to use both statistical measures.

10)  Discussion reads well, but can be further expanded on the implications of the findings, particularly in future policy to manage BE cases. Are there any recommendations, or implications, from the findings of this study? How are the study findings important?

Responses. Thanks for raising this important point. We have added in the discussion section that the results obtained in this study may be important in the management of patients with BE, in favor of certain recommendations to minimize exposure to respiratory germs between another recommendations.

Reviewer 3 Report

Martinez-Vergara and colleagues report an observational, retrospective- prospective single center study, exploring various aspects of bronchiectasis in a year before the COVID19 pandemic and during the first year of the pandemic. They have found a significant decrease in pulmonary exacerbations during the pandemic, although symptom scores did not change. Moreover, patients gave less sputum samples, and surprisingly, the rate of most potentially pathogentic microorganisms (PPM)- total, PA, HI- significantly decreased.

What is very striking to me, is the finding of a decrease in PPM in sputum cultures. The authors suggest an explanation: less viral- induced exacerbations causing “bacterial superinfection”. I would argue against the term “superinfection” which implies an acute bacterial infection when, in fact, several studies have shown that there is very little difference between stable and exacerbation airway microbiome. Another explanation may be that the “threshold” quantity of bacteria allowing detection in cultures (and reporting by the lab) is exceeded more often in the pre-pandemic era due to more frequent exacerbations. It would be therefore useful to analyze how many of the cultures in both periods were during stable vs. exacerbation phase, and whether the drop in PPM was driven by a decrease in PPM in one of these conditions (stable vs. exacerbations).

The paper is well written, and the methodology is sound. My major comment is that it could be shortened- there is repeated presentation of what is basically the same finding- a decrease in exacerbations (The improvement in BSI and E-FACED is not surprising, since both scores are heavily influenced by exacerbations and presence of bacteria).

Minor comments-

Abstract- it unclear what a “10%” reduction in PA (for example) mean, and preferable to write the mean values in the two periods.

Table 1- “cycles”- term is unclear . Do the authors mean antibiotic courses?

Line 205: The percentage of patients from whom it was not possible to collect valid sputum samples was greater in the pre-pandemic period, with a mean (range) of 2.5 (0-9), as opposed to 1.2 (0-6) during the pandemic; p=0.012. What is the significance of this finding?

Author Response

Martinez-Vergara and colleagues report an observational, retrospective- prospective single center study, exploring various aspects of bronchiectasis in a year before the COVID19 pandemic and during the first year of the pandemic. They have found a significant decrease in pulmonary exacerbations during the pandemic, although symptom scores did not change. Moreover, patients gave less sputum samples, and surprisingly, the rate of most potentially pathogenic microorganisms (PPM)- total, PA, HI- significantly decreased.

What is very striking to me, is the finding of a decrease in PPM in sputum cultures. The authors suggest an explanation: less viral- induced exacerbations causing “bacterial superinfection”. I would argue against the term “superinfection” which implies an acute bacterial infection when, in fact, several studies have shown that there is very little difference between stable and exacerbation airway microbiome. Another explanation may be that the “threshold” quantity of bacteria allowing detection in cultures (and reporting by the lab) is exceeded more often in the pre-pandemic era due to more frequent exacerbations. It would be therefore useful to analyze how many of the cultures in both periods were during stable vs. exacerbation phase, and whether the drop in PPM was driven by a decrease in PPM in one of these conditions (stable vs. exacerbations).

Response. This is an interesting point and the hypotheses exposed by the reviewer are some possibilities. Unfortunately we have not the data of quantitative cultures. Culture results were expressed as CFU per milliliter and ≥10^3 CFU were considered positive for a given pathogenic microorganisms (as in all the manuscripts derived from our registry). We have added this information now in the methodology of the paper and as a limitation of the study.  Finally, beyond the number of collected sputum sample per patient, we have not information about the percentage of sputum samples during an exacerbation and in stable phase of the disease, so we cannot make analysis about the interesting proposal of the reviewer, and as a consequence we add this hypothesis in the text (discussion section) and also our limitation.

The paper is well written, and the methodology is sound. My major comment is that it could be shortened- there is repeated presentation of what is basically the same finding- a decrease in exacerbations (The improvement in BSI and E-FACED is not surprising, since both scores are heavily influenced by exacerbations and presence of bacteria).  

Response: We have shorten some repeated sentences in the text. On the other hand, as the reviewer says, “the improvement in BSI and E-FACED is not surprising since both scores are heavily influenced by exacerbations and presence of bacteria”. This is absolutely true, however, we think that the inclusion of both severity bronchiectasis scores is important in the context of this paper.

Minor comments-

Abstract- it unclear what a “10%” reduction in PA (for example) mean, and preferable to write the mean values in the two periods.

Response: Thanks for the correction. We have added the mean values in the two periods in abstract.

Table 1- “cycles”- term is unclear. Do the authors mean antibiotic courses?

Response: Thanks for the correction, Yes, it was a translation mistake.  We have replaced “cycles” for “courses”

Line 205: The percentage of patients from whom it was not possible to collect valid sputum samples was greater in the pre-pandemic period, with a mean (range) of 2.5 (0-9), as opposed to 1.2 (0-6) during the pandemic; p=0.012. What is the significance of this finding?

Response. Again, it was a mistake. Thanks for the comment. We have now amended it.